# MBLinhibitors.com, a Website Resource Offering Information and Expertise for the Continued Development of Metallo-β-Lactamase Inhibitors

**DOI:** 10.3390/biom10030459

**Published:** 2020-03-16

**Authors:** Zishuo Cheng, Caitlyn A. Thomas, Adam R. Joyner, Robert L. Kimble, Aidan M. Sturgill, Nhu-Y Tran, Maya R. Vulcan, Spencer A. Klinsky, Diego J. Orea, Cody R. Platt, Fanpu Cao, Bo Li, Qilin Yang, Cole J. Yurkiewicz, Walter Fast, Michael W. Crowder

**Affiliations:** 1Department of Chemistry and Biochemistry, Miami University, Oxford, OH 45056, USA; chengz5@miamioh.edu (Z.C.); thomas60@miamioh.edu (C.A.T.); kimblerl@miamioh.edu (R.L.K.); sturgiam@miamioh.edu (A.M.S.); trann@miamioh.edu (N.-Y.T.); vulcanmr@miamioh.edu (M.R.V.); klinsksa@miamioh.edu (S.A.K.); orea001@cougars.csusm.edu (D.J.O.);; 2Department of Computer Science and Software Engineering, Miami University, Oxford, OH 45056, USA; joynerar@miamioh.edu (A.R.J.); plattcr@miamioh.edu (C.R.P.);; 3Division of Chemical Biology and Medicinal Chemistry, College of Pharmacy and the LaMontagne Center for Infectious Disease, University of Texas, Austin, TX 78712, USA

**Keywords:** antibiotic resistance, metallo-β-lactamase, website, inhibitor, mblinhibitor.com

## Abstract

In an effort to facilitate the discovery of new, improved inhibitors of the metallo-β-lactamases (MBLs), a new, interactive website called MBLinhibitors.com was developed. Despite considerable efforts from the science community, there are no clinical inhibitors of the MBLs, which are now produced by human pathogens. The website, MBLinhibitors.com, contains a searchable database of known MBL inhibitors, and inhibitors can be searched by chemical name, chemical formula, chemical structure, Simplified Molecular-Input Line-Entry System (SMILES) format, and by the MBL on which studies were conducted. The site will also highlight a “MBL Inhibitor of the Month”, and researchers are invited to submit compounds for this feature. Importantly, MBLinhibitors.com was designed to encourage collaboration, and researchers are invited to submit their new compounds, using the “Submit” function on the site, as well as their expertise using the “Collaboration” function. The intention is for this site to be interactive, and the site will be improved in the future as researchers use the site and suggest improvements. It is hoped that MBLinhibitors.com will serve as the one-stop site for any important information on MBL inhibitors and will aid in the discovery of a clinically useful MBL inhibitor.

## 1. Introduction

Antibiotic resistance is becoming an increasingly important biomedical issue, turning what was once easily treated with inexpensive and easily-accessible antibiotics into untreatable infections [1]. According to the Centers for Disease Control and Prevention (CDC), 2.8 million infections occur from antibiotic-resistant bacteria in the U.S. each year, with about 35,000 deaths from these infections [2]. The World Health Organization (WHO) predicts that over 10 million deaths, as well as an economic loss of $10 trillion, will occur annually if effective intervention is not implemented [3]. Since the discovery of penicillin by Fleming in 1929, the β-lactam class remains the largest class of antibiotics for the treatment of bacterial infections, making up 65% of the antibacterial arsenal [4]. Penicillins, cephalosporins, carbapenems, and monobactams are all members of the β-lactam class [5]. The widespread use of this class of antibiotics has led to the emergence of different resistance mechanisms, including: (a) the production of altered penicillin binding proteins (PBP) with lower binding affinities for most β-lactam antibiotics; and (b) the production of β-lactamases, which is the most common resistance mechanism in Gram-negative bacteria [6]. In 2019, there are more than 2800 identified β-lactamase genes [7]. They have been classified biochemically into two categories according to the mechanism by which they hydrolyze the β-lactam bond [8]. The serine-β-lactamases (SBL) utilize an active site serine to hydrolyze the β-lactam bond. The metallo-β-lactamases (MBL) utilize Zn(II)-containing active sites to hydrolyze the β-lactam bond in these antibiotics [9].

Although the SBLs are more prevalent in the clinic over the past seventy years, there exist inhibitors, which can be given in combination with other β-lactam containing antibiotics, to treat bacteria that produce some of the SBLs [10]. Examples of these FDA-approved inhibitors include clavulanic acid, sulbactam, avibactam, and tazobactam [10]. However, despite considerable efforts to develop such inhibitors [6], there are no clinically-approved inhibitors that are available for MBLs, making infections from bacteria that produce MBL a serious challenge. An ideal MBL inhibitor would have good inhibition properties, low toxicity, and is broad-spectrum [11]. Three major challenges have limited success in preparing a clinical inhibitor of the MBLs. Firstly, there are large structural variances exhibited by the MBLs, even those from the same molecular subclass [12]. There are three subclasses of MBLs; B1, B2, and B3, and members are distinguished by amino acid sequence, molecular properties, identity of Zn(II)-coordinating ligands, and the number of active site metal ions present [9]. Across these subclasses, there is less than 20% amino acid sequence identities [13]. In the B1 subclass alone, there is only a modest 30% amino acid sequence similarities, with only a few highly-conserved residues present outside the Zn(II)-binding site [12]. This structural diversity has resulted in MBL inhibitors that inhibit only one (or a few) MBL, but not others. For example, the dicarboxylic acid compound ME1071 was reported to be a good inhibitor of MBL IMiPenemase (IMP-1) and VIM-2 Verona Integron-borne MBL (VIM-2) [14]. However, this compound is a poor inhibitor of subclass B1 MBL NDM-1 New Delhi MBL (NDM)-1) [15]. Another example is the bicyclic boronate VNRX-5133, which exhibits good inhibition against NDM and other subclass B1 enzymes [16]; however, this compound is not a good inhibitor of subclass B3 MBL L1 [16]. Secondly, it is imperative that any clinical MBL inhibitor be selective towards bacterial MBLs over human MBL-fold containing enzymes, some of which have important physiological roles [6]. The most common (and perhaps most obvious) way to inhibit an MBL is through the use of a chelating agent that binds to the Zn(II) ion(s) in the active site [17]. One can envision two limiting inhibition mechanisms used by such inhibitors: (1) stripping of the Zn(II) from the active site; or (2) coordination of the Zn(II) ion(s) while they are bound to the MBL to produce a ternary complex [17]. Many inhibitors that have metal binding scaffolds exhibit good in vitro inhibition properties [17], and one natural product, aspergillomarasmine (AMA), was reported to be effective in an in vivo infection model [18]. However, subsequent studies revealed unacceptable toxicities (50% lethal dose [LD_50_] in mice is 159.8 mg/kg), which limited its further development [19]. This study and others have suggested that metal-targeting inhibitors will need to be developed that form ternary complexes [17]. Thirdly, the continued emergence of new MBL variants exceeds the rate at which new inhibitors are currently being developed. The IMP-type MBLs were identified first in the late 1980s [20], while the VIM-type MLBs were first discovered in 1995 [21]. NDM-1 was first detected in 2008 [22]. There are currently 29, 69, and 85 clinical variants of NDM, VIM, and IMP listed in the β-lactamase database, respectively [23], and the number of clinical variants will continue to grow (Figure 1A). There is growing evidence that MBL inhibitors interact differently with different variants [24,25]. For example, one isoquinoline derivative potently inhibited VIM-5 and VIM-38, while the same compound was not nearly as potent toward VIM-1, VIM-2, or VIM-4 [26].

In the past five years, there has been significant effort to identify novel inhibitors of the MBL [6,11]. The numbers of MBL inhibitors and publications describing MBL inhibitors have increased over the last decade (Figure 1B and Figure 1C). The four major approaches to identify MBL inhibitors include: (1) high-throughput screening (HTS) of large chemical libraries [18]; (2) fragment-based drug discovery (FBDD) [27,28]; (3) computer-based virtual screening [29,30]; and (4) screening of natural products [31]. The typical “hit” rates of these approaches are very low (<1%) [32]. Despite the low success rates, over 900 inhibitors have been reported in literature as of October 2019. Given this number of compounds, there have been several review articles that have categorized the inhibitors based on structure [33,34] or mechanism of inhibition [12,17]. While these articles provide excellent insights to existing MBL inhibitors, and often suggest future directions, most review articles focus on the best inhibitors and do not often discuss compounds with relatively poor inhibition properties, even though these latter compounds may be great scaffolds for redesign efforts. In addition, review articles are typically not up to date, and some of them are published in journals that are difficult to access. With the increasing number of compounds (Figure 1C), it is becoming more difficult to collect and sort through the review articles or the original articles, which are not interactive or searchable.

To address these issues, we have developed a website called MBLinhibitors.com (link: https://MBLinhibitors.miamioh.edu), which is the first effort, to our knowledge, to provide a searchable database of current and future MBL inhibitors. The primary goal of the website is to provide up-to-date information on MBL inhibitors to facilitate current and future MBL inhibitor development and/or redesign efforts. The initial form of MBLinhibitors.com includes structures of the MBL inhibitors, SMILES formats, toxicity results if available, mechanism of inhibition if known, IC_50_ values if reported, targeted enzymes, and primary references. It is hoped that additional information will be included as researchers start to use the website and offer suggestions and data to improve the site. The site has been designed for researchers to add their new inhibitors, and we hope that researchers will submit compounds with great promise, but also compounds that are not as promising at this time. The information about these latter compounds might prevent researchers from testing compounds that have already been shown to be poor MBL inhibitors, and ultimately save time and money to work on more promising candidates. The overarching hope is that providing up-to-date information in a concise, organized manner, and more importantly in one location, will facilitate more rapid development of future MBL inhibitors, and discovery of clinically useful compounds.

## 2. Materials and Methods

### 2.1. Data Collection

Literature searches (Keywords include metallo-beta-lactamase and inhibitor) were conducted using PubMed [35] and Google Scholar [36]. Articles reporting MBL inhibitors were selected manually. The structures of compounds were obtained from the literature and drawn using ChemDraw Version 18.1 (PerkinElmer, Waltham, MA, USA) In vitro inhibition (K_i_ and IC_50_ values), in vivo (minimum inhibitory concentration (MIC)), toxicity, inhibition, and crystal structural data were obtained from the articles. SMILES formats, molecular weights, and molecular formulas were determined using ChemDraw. All data were compiled in Microsoft Excel spreadsheets and converted into a comma-separated values (CSV) file.

### 2.2. Website Construction

In order to minimize running costs and retain ease of maintenance, a LAMP (Linux, Apache, MySQL, and Personal Home Page (PHP) stack was used to design the site. For development, PHP version 7.2 (Zend Technologies, Cupertino, CA, USA) and MySQL version 8.0.18 (Oracle, Redwood City, CA, USA) were used. The website was developed using Visual Studio Code, developed by Microsoft (Redmond, WA, USA). For the front-end, a framework called ‘Materialize’ was used in order to design a clean, uniform interface. All SMILES format drawings were dynamically drawn using a JavaScript library (Oracle, Redwood City, CA, USA) named ‘SmilesDrawer’ that was developed by Daniel Probst and Jean-Louis Reymond [37] (Berne, Switzerland). Any database calls were made by making HTTP requests to an internal PHP Application Programming Interface (API). The API parsed the incoming GET and POST requests and was able to connect to the MySQL database using the MySQLi class. The API could be exposed in the future to enable others to use this website’s data in their own applications.

## 3. Results

The home page of the website includes a short video by Michael Crowder, which describes the website and the rationale for developing MBLinhibitors.com (Figure 2). The home page also includes a short feature called the “MBL Inhibitor of the Month.” For the launch of the website, the developers chose to highlight recent work from Professor Chris Schofield’s group at the University of Oxford (Oxford, UK) regarding his bicyclic boronate (VNRX-5133, Venatorx Pharmaceuticals, Malvern, PA, USA) inhibitor [16]. Professor Schofield and coworkers wrote a short perspective of the compound, and it is the intent to highlight new compounds each month. We will invite researchers to submit their compounds and descriptions of the compounds.

The home page also contains four menus (Figure 2), in addition to a home button. These menus are:
“Background”: This menu leads to information about the various MBLs and MBL inhibitors included within the website.“Database”: This menu contains two functions; “Keyword Searching” and “Functional Group Filter.” The former allows visitors to search for a specific MBL inhibitor by chemical name, chemical formula, chemical structure, and/or SMILES format (Figure 3). The latter allows visitors to narrow their search results based on one or more functional groups that make up the inhibitor. Researchers can also search by specific MBL, by simply the name of the enzyme (NDM, VIM, etc.). Once a search has been completed, a list of the inhibitors, with the name and chemical formula, is shown (Figure 3). By clicking on the “View” button, the researcher can access a structure of the compound, name, SMILES, molecular weight, testing data (if known), and a link to a relevant publication.“Collaboration”: This menu currently contains a link to a description of Michael Crowder’s research lab, email address, link to website, and a description of techniques conducted in the lab on MBL inhibitors. To encourage collaboration in the MBL community, a second link is an invitation to other researchers to join the list of collaborators, with a fillable form to provide information. Researchers can elect to submit information that will be posted on MBLinhibitors.com.“Submit”: This function allows for researchers to submit new compounds to the database. Researchers can submit the chemical name, SMILES format (after this information is submitted, a structure is automatically drawn in the window), chemical formula, molecular weight, and publication information. After submission, the compound information will be reviewed and added to the mblinhibitor.com database.

## 4. Discussion

### 4.1. Data Sharing

On this website, visitors have access to information on nearly 1000 different compounds, and compounds can be searched using the Inhibitor Database tab by chemical name, chemical formula, chemical structure, SMILES format, or by using the functional group filter. More detailed information about compounds in the database can be obtained by clicking the PubMed ID link associated with the compound. When developing this database, we chose information on the MBL inhibitors that we believed to be most useful to other research groups; however, we encourage researchers to contact us (using the Contact link on the Home page) about including additional data in the database. Our group will continue to conduct literature searches to find newly-reported MBL inhibitors or new inhibition/toxicity/in vivo/structural data on previously-reported compounds. However, researchers will be encouraged to submit their new compounds and data directly to the website, using the submit function under the “Submit” menu. The new compounds would include both compounds that are good inhibitors, and not good inhibitors. As we discussed in the Introduction section, the success rate for identifying new inhibitors from screening large chemical compounds is only 1%, and the success rate for modified scaffolds is 12%–80% [32]. Therefore, there is a lot of unreported information in these screens that could be used to guide future structure/activity efforts. We hope that this website will be a repository of all MBL inhibitor discovery efforts, including the large portion that is now invisible to the MBL community.

In addition, we believe the information in this website has great potential in the future for in silico analyses. Drug discovery is resource-intensive, and it is estimated to cost drug makers $2.6 billion and 10–20 years to develop new prescription medicines [38]. Artificial intelligence (AI) may be used to accelerate this process, and reduce costs by more quickly identifying promising leads [39]. Even though no MBL inhibitors have been discovered by using machine learning so far, this approach has been used to identify inhibitors of other biomedically-important enzymes [40,41]. Each compound in our database includes its SMILES format, which is commonly used in in silico analyses.

### 4.2. Expertise Sharing

Based on literature searches and analyses of resulting data that we compiled on 930 compounds, 100% of the compounds have some steady state inhibition data, 36% have reported MIC data, 5% have reported mechanisms of inhibition, 12% have toxicity data, and 4% have crystal structures with MBL inhibitor(s) complexed to a MBL. Even though a goal of most MBL inhibitor discovery efforts is finding a pan inhibitor, only 22% of the compounds were tested against MBLs from multiple subclasses, and only one of the 930 compounds was tested against variants of a single MBL. Given the number of current compounds with missing data, the increase in number of new compounds being reported, and the increase in number of clinical variants [23], it will be necessary for researchers to share expertise and work together to fill the information void. We hope that MBL researchers will be willing to post their names and contact information on the site, and details about their research programs; interested researchers should contact the Crowder group.

## 5. Conclusions

In this paper, we describe a new, interactive website called MBLinhibitors.com. We believe it is an open-access, one-stop solution for the developers of MBL inhibitors, to get the most up-to-date information and to facilitate the formation of collaborations.

## Figures and Tables

**Figure 1 biomolecules-10-00459-f001:**
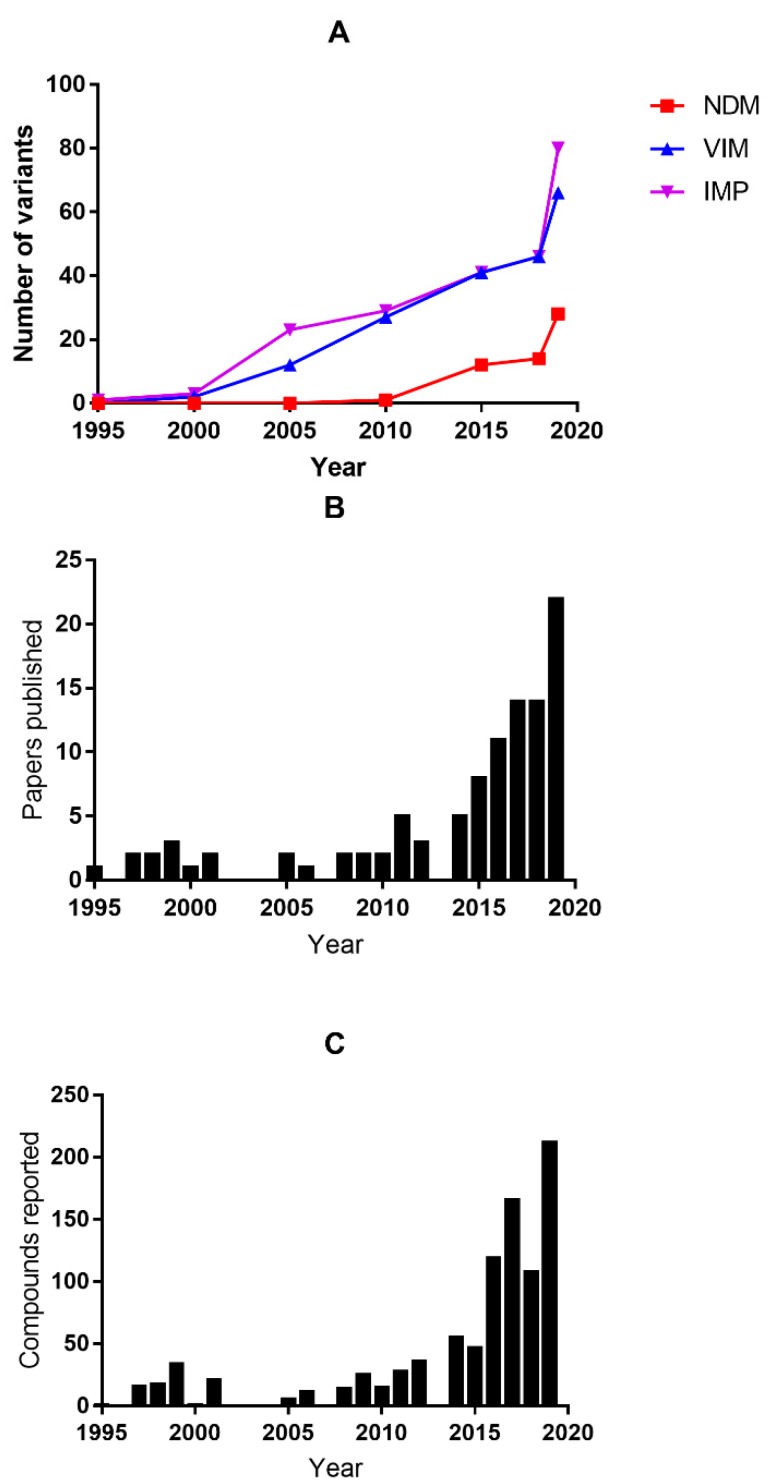
Increase in metallo-β-lactamases (MBL) and MBL inhibitors. (**A**) Selective MBL enumerated according to http://bldb.eu/. (**B**) Number of publications related to MBL inhibitors from year 1995 to 2019. (**C**) Number of reported MBL inhibitor from year 1995 to 2019.

**Figure 2 biomolecules-10-00459-f002:**
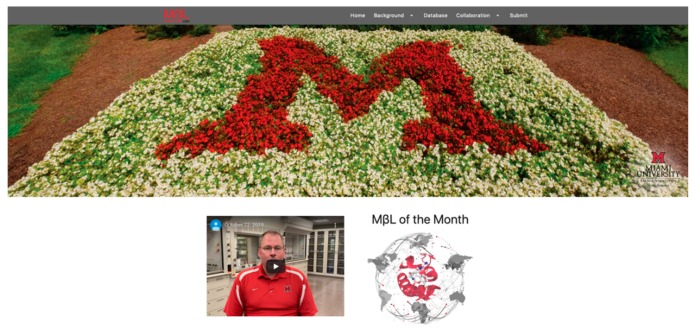
The home page of MBLinhibitor.com, which includes the main buttons, a short introductory video, and a short feature called the “MBL Inhibitor of the Month.

**Figure 3 biomolecules-10-00459-f003:**
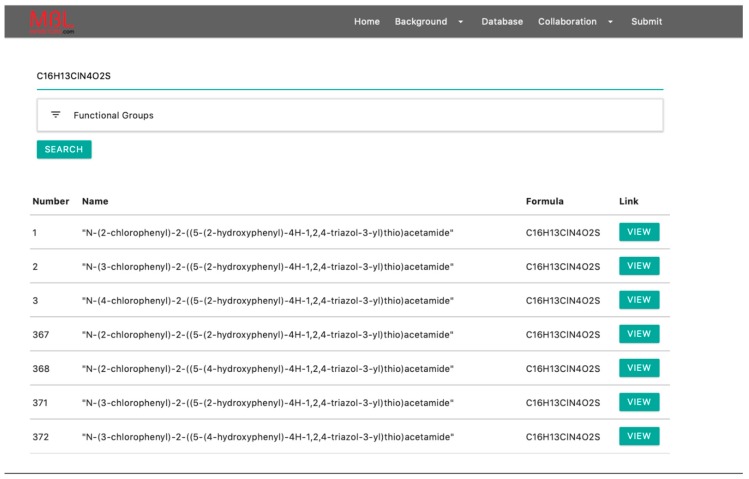
The demo of formula searching, using C16H13ClN4O2S as the keyword. More detailed info can be obtained by clicking the “VIEW” button for each compound.

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
