# Peer review of "MBLinhibitors.com, a Website Resource Offering Information and Expertise for the Continued Development of Metallo-β-Lactamase Inhibitors"

_biomolecules, 2020, doi:10.3390/biom10030459_

Round 1

Reviewer 1 Report

The manuscript of Cheng and coworkers (biomolecules-735859) introduces to the research community a novel website that aims at becoming a point of reference for the research on metallo-beta-lactamase (MBL) inhibitors. The importance of such inhibition in the field of antibiotic resistance is well known, and accurately explained in the Introduction. The rest of the manuscript describes the collection of the data (the compounds tested, their molecular features, their efficacy as inhibitors, toxicity, etc.), and how the website was built and organized (including the possibility for the users to add new data, and to find collaborations on common research projects).

I have browsed for a little while on the website (https://mblinhibitors.miamioh.edu), and I found it very nice, including the introductory video. I few nitpicks I would like to report are the following: (1) too big fonts are used in too many places; (2) the mail address [email protected] sounds not much professional; (3) the red logo on top left is very pixelated. In any case, these are just minor issues, and they do not affect the usability of the site. On the other hand, I believe that the manuscript has some flaws (a single major one, and a number of minor issues) that need to be addressed before being published on Biomolecules.

Major point:

I found there is a great confusion in the name (and possibly in the internet address) used for the website. Name and address of a site are distinct entities, and both are (separately) case sensitive. Best practices suggest that a mix of uppercase and lowercase letters should be used for the former, and only lowercase letters for the latter (e.g., the site UniProt at www.uniprot.org). So the first issue is about the name of the site: is it “MBLinhibitors.com”, as stated in the title, or “mblinhibitors.com”, as it is called everywhere else along the manuscript?

And the second issue is: what is the purpose of the “.com” in the title, when it is nowhere to be found in the address? The “.com” suffix makes sense only if the site is “www.mblinhibitors.com”; and in fact I immediately tried to go to that address, with no success. It is important to note that the actual address https://mblinhibitors.miamioh.edu is only mentioned much later in the manuscript (at the end of page 3, and neither in the Abstract nor in the Conclusions), which also suggests to cite it earlier and in perhaps in more relevant sections of the manuscript.

Minor points:

Line 14: “metallo-beta-lactamases (MBL)”. If the definition includes the plural (“lactamases”), then later it cannot be used “MBLs” (e.g., line 52, 55, etc.).

Line 24: “iterative”. It is probably meant “interactive”.

Line 34: “WHO (World Health Organization)”. The full name should normally precede the acronym. The same applies to lines 63-65.

Line 48-50: This sentence is somewhat garbled.

Line 55: “stymied”. This terms sounds a bit informal.

Line 81-82: It would be better do add “respectively” somewhere in the sentence.

Line 88: “The numbers of MBL inhibitors and publications describing MBL inhibitors have increased dramatically”. In the case of publications, as it is clear from the Figure, “dramatically” may be excessive.

Line 125: “Google scholar”. It should be “Scholar”.

Line 130: “All data were compiled in Microsoft Excel spreadsheets”. This detail is irrelevant.

Line 139: Various acronyms are used, but maybe at least “Application Programming Interface” (API) should be defined.

Line 151: “The home page also contains four drop down menus (Figure 2)”. Technically speaking, there are only two drop down menus (those with the arrow pointing downwards).

Line 156 and following: The bullet points and the text alignment in the description of the menus should be uniformed. The same attention in the pagination of the manuscript should be paid on line 166.

Line 164: “corresponding to the search”. This is quite obvious.

Line 190: “the introduction”. I would suggest “the Introduction section”.

Line 206: “SMILES (“Simplified Molecular-Input Line-Entry System”)”. It is ironic that this acronym is present several times and defined only the very last time it is used.

Line 214: “only one of the 930 compounds were tested”. It should be “was”.

Line 222: “open-access”. This is an important information, and should be mentioned somewhere before in the manuscript.

Finally, I would recommend to improve also the References section. For instance, ref. 22 has many spurious HTML tags, ref. 35 contains the journal abbreviation (at variance with the complete name used elsewhere), and in many reference the term “beta” should be substituted with the Greek letter.

Author Response

Reviewer 1: 

I have browsed for a little while on the website (https://mblinhibitors.miamioh.edu), and I found it very nice, including the introductory video. I few nitpicks I would like to report are the following: (1) too big fonts are used in too many places;

We have carefully gone through the website, and used computer science majors to review the page, and suggest changes using standard and current practice in website design. We didn’t make many changes based on this analysis.

(2) the mail address [email protected] sounds not much professional;

We have removed this email address in the revised website and replaced it with my email address: [email protected]. This suggestion and change will ensure that we address email messages promptly, and we appreciate this suggestion from the reviewer.

(3) the red logo on top left is very pixelated. In any case, these are just minor issues, and they do not affect the usability of the site.

We have replaced the initial logo with an improved, higher resolution logo.

On the other hand, I believe that the manuscript has some flaws (a single major one, and a number of minor issues) that need to be addressed before being published on Biomolecules.

Major point:

I found there is a great confusion in the name (and possibly in the internet address) used for the website. Name and address of a site are distinct entities, and both are (separately) case sensitive. Best practices suggest that a mix of uppercase and lowercase letters should be used for the former, and only lowercase letters for the latter (e.g., the site UniProt at www.uniprot.org). So the first issue is about the name of the site: is it “MBLinhibitors.com”, as stated in the title, or “mblinhibitors.com”, as it is called everywhere else along the manuscript?

We have corrected this issue. We have changed all “mblinhibitors.com” to “MBLinhibitors.com”. It is important to note that a web search is case insensitive. Either name points to the exact same site.

And the second issue is: what is the purpose of the “.com” in the title, when it is nowhere to be found in the address? The “.com” suffix makes sense only if the site is “www.mblinhibitors.com”; and in fact I immediately tried to go to that address, with no success. It is important to note that the actual address https://mblinhibitors.miamioh.edu is only mentioned much later in the manuscript (at the end of page 3, and neither in the Abstract nor in the Conclusions), which also suggests to cite it earlier and in perhaps in more relevant sections of the manuscript.

We have fixed this issue, and the official and searchable name of the website is MBLinhibitors.com. We have tested this domain name in Safari, Chrome, and Firefox. We apologize for the confusion that our submitted paper caused for the reviewer. For an explanation, we initially set up the website on a commercial site called WIX. For security and stability issues, we transferred the site to a Miami University server, which requires that we used the *.miamioh.edu ending to the website. We believe that MBLinhibitors.com will be an easy address for users to use. So, we used the Hover website to transfer and rename the domain name. When we submitted the manuscript for review, we were waiting for Hover to complete our request. Hover has completed the request and our website is fully searchable now, using MBLinhibitors.com.

Minor points:

Line 14: “metallo-beta-lactamases (MBL)”. If the definition includes the plural (“lactamases”), then later it cannot be used “MBLs” (e.g., line 52, 55, etc.).

 We included an “s” in the abbreviation defined in the Abstract. Then it is clear later in the manuscript that “MBL” refers to “metallo--lactamase” and “MBLs” refers to “metallo--lactamases”

Line 24: “iterative”. It is probably meant “interactive”.

 The reviewer is correct. We have changed the word “iterative” to “interactive”

Line 34: “WHO (World Health Organization)”. The full name should normally precede the acronym. The same applies to lines 63-65.

In the revised manuscript, we have moved “World Health Organization” on line 35 in front of “(WHO)” as suggested by the reviewer.

The line numbering in the manuscript has changed, but we have moved the names of the MBLs in front of their abbreviations in lines 68-70 in the revised manuscript. The sentence now reads, “For example, the dicarboxylic acid compound ME1071 was reported to be a good inhibitor of MBL IMiPenemase (IMP-1) and VIM-2 Verona IMipenemase (VIM-2) [14]. However, this compound is a poor inhibitor of subclass B1 MBL NDM-1 New Delhi MBL (NDM)-1) [15].”

Line 48-50: This sentence is somewhat garbled.

Again, the line numbers are different, but we believe that the reviewer is referring to the first sentence in the first full paragraph on page 2. We left out the word “which” in the sentence. The new sentence reads, “Although the SBLs are more prevalent in the clinic over the past seventy years, there exists inhibitors, which can be given in combination with other -lactam containing antibiotics, to treat bacteria that produce some of the SBLs [10].”.

Line 55: “stymied”. This terms sounds a bit informal.

We changed “stymied” to “limited”. The new sentence (on line 60) now reads, “Three major challenges have limited success in preparing a clinical inhibitor of the MBLs.”.

Line 81-82: It would be better do add “respectively” somewhere in the sentence.

We added the word “respectively” at the appropriate place. The new sentence (line 87) now reads, “There are currently 28, 66, and 80 clinical variants of NDM, VIM, and IMP listed in the -lactamase database, respectively [23], and the number of clinical variants will continue to grow (Figure 1A).”.

Line 88: “The numbers of MBL inhibitors and publications describing MBL inhibitors have increased dramatically”. In the case of publications, as it is clear from the Figure, “dramatically” may be excessive.

We disagree with the reviewer; the increases are nearly exponential. However, we have removed the offending word “dramatically” in the revised manuscript. The new sentence (lines 94-95) now reads, “The numbers of MBL inhibitors and publications describing MBL inhibitors have increased over the last decade (Figure 1B and 1C).”.

Line 125: “Google scholar”. It should be “Scholar”.

We have changed “Google scholar” to “Google Scholar” on line 135.

Line 130: “All data were compiled in Microsoft Excel spreadsheets”. This detail is irrelevant.

We believe that this information is important. It reveals to the readers how we compiled (and coverted) the data. For anyone wanting to use the database for future machine learning/AI work (we are currently doing this), it is helpful to know how the data were compiled. We prefer not to delete this sentence.

Line 139: Various acronyms are used, but maybe at least “Application Programming Interface” (API) should be defined.

We added the definition of “API” at the appropriate place (line 151), and the definition of PHP (Personal Home Page) at the appropriate place (line 155). GET and POST are names of HTTP, and these are not abbreviations.

Line 151: “The home page also contains four drop down menus (Figure 2)”. Technically speaking, there are only two drop down menus (those with the arrow pointing downwards).

We removed “drop down” from the sentence. The new sentence (line 162) reads, “The home page also contains four menus (Figure 2), in addition to a home button.”.

Line 156 and following: The bullet points and the text alignment in the description of the menus should be uniformed. The same attention in the pagination of the manuscript should be paid on line 166.

We believe that we fixed this problem, which was that the last bulleted item was indented one too many times. We believe that they are all aligned correctly in the revised manuscript (see page 4; bulleted items).

Line 164: “corresponding to the search”. This is quite obvious.

We removed the phrase, “corresponding to the search”. The new sentence (lines 171-172) now reads, “Once a search has been completed, a list of the inhibitors, with the name and chemical formula, is shown (Figure 3).”.

Line 190: “the introduction”. I would suggest “the Introduction section”.

We added the word “section” into the sentence. The new sentence (line 199) now reads, “As we discussed in the Introduction section, the success rate for identifying new inhibitors from screening large chemical compounds is only 1%, and the success rate for modified scaffolds is 12-80% [32].”.

Line 206: “SMILES (“Simplified Molecular-Input Line-Entry System”)”. It is ironic that this acronym is present several times and defined only the very last time it is used.

In the revised manuscript, we have now defined SMILES in the Abstract, which is the first place that it is mentioned. We removed the definition from the end of the manuscript.

Line 214: “only one of the 930 compounds were tested”. It should be “was”.

We changed “were” to “was”. The new sentence (line 225) now reads, “only 22% of the compounds were tested against MBLs from multiple subclasses, and only one of the 930 compounds was tested against variants of a single MBL.”.

Line 222: “open-access”. This is an important information, and should be mentioned somewhere before in the manuscript.

We don’t understand this comment. We never mention in the manuscript that this is fee-based website; in fact, we have provided the weblink. We believe that it is clear that our website if a free, open-access website throughout the paper.

Finally, I would recommend to improve also the References section. For instance, ref. 22 has many spurious HTML tags, ref. 35 contains the journal abbreviation (at variance with the complete name used elsewhere), and in many reference the term “beta” should be substituted with the Greek letter.

We removed HTML tags from ref. 22. We applied NCBI NLM title abbreviation in all journal names. We substituted the term “beta” with the Greek letter β except for ref 23, which “beta” was used in the original paper title.

Reviewer 2 Report

Dear Authors,

this is a very interesting communication about a topic that interests many practitioners and researchers in the field of human and veterinary medicine.

I think that "mblinhibitors.com" will be a helpful tool for many people who work in this field.

best regards

Author Response

There were no criticisms from Reviewer #2 to which to respond.

Reviewer 3 Report

Antibiotic resistance is one of the major Public Health issues of the century and new therapeutic options are needed. This manuscript refers to the construction of a website with a database that aims at gathering information on tested MBL inhibitors. It seems a good idea, especially the inclusion of inhibitors that do not have good activity, as traits of these are often not evidenced in publications and are good starting points for further work, as well as prevent others from wasting time testing the same compounds. It is a well written and organized manuscript.

A few minor comments that should be revised:

Line 37: I think authors wanted to say “treatment of bacterial infection” rather than “treatment of bacterial resistance”.

Line 104: Figure 1, the first “A” in the legend should be deleted. “Increase in MBLs and MBL inhibitors” is the legend of the entire figure, not only from graphic A.

Line 158: A bullet point is missing before “Database”.

Line 206: The meaning of SMILES should be introduced in the text where the word is mentioned for the first time (line 112).

Author Response

Reviewer 3:

A few minor comments that should be revised:

Line 37: I think authors wanted to say “treatment of bacterial infection” rather than “treatment of bacterial resistance”.

We have changed the word “resistance” to “infections”. The new sentence now reads, “Since the discovery of penicillin by Fleming in 1929, the β-lactam class remains the largest class of antibiotics for the treatment of bacterial infections, making up 65% of the antibacterial arsenal [4].”.

Line 104: Figure 1, the first “A” in the legend should be deleted. “Increase in MBLs and MBL inhibitors” is the legend of the entire figure, not only from graphic A.

We removed the “A.” from the Figure 1 legend, as recommended by the reviewer.

Line 158: A bullet point is missing before “Database”.

We fixed this missing bullet point on line 166.

Line 206: The meaning of SMILES should be introduced in the text where the word is mentioned for the first time (line 112).

We inserted the meaning of SMILES in the Abstract, where it was first mentioned.